# In Silico Approach of Glycinin and Conglycinin Chains of Soybean By-Product (Okara) Using Papain and Bromelain

**DOI:** 10.3390/molecules27206855

**Published:** 2022-10-13

**Authors:** Andriati Ningrum, Dian Wahyu Wardani, Nurul Vanidia, Achmat Sarifudin, Rima Kumalasari, Riyanti Ekafitri, Dita Kristanti, Woro Setiaboma, Heli Siti Halimatul Munawaroh

**Affiliations:** 1Department of Food and Agricultural Product Technology, Faculty of Agricultural Technology, Universitas Gadjah Mada, Flora Street No. 1, Bulaksumur, Yogyakarta 55281, Indonesia; 2Research Centre for Appropriate Technology, National Research and Innovation Agency, KS. Tubun Street No.5, Subang 41213, Indonesia; 3Research Center for Food Technology and Processing, National Research and Innovation Agency, Jogja-Wonosari Street km 31,5 Playen, Gunungkidul, Yogyakarta 55861, Indonesia; 4Study Program of Chemistry, Department of Chemistry Education, Faculty of Mathematics and Science Education, Universitas Pendidikan Indonesia, Bandung 40154, Indonesia

**Keywords:** antioxidant, bioactive peptides, okara, in silico, valorization

## Abstract

This study explores utilization of a sustainable soybean by-product (okara) based on in silico approach. In silico approaches, as well as the BIOPEP database, PeptideRanker database, Peptide Calculator database (Pepcalc), ToxinPred database, and AllerTop database, were employed to evaluate the potential of glycinin and conglycinin derived peptides as a potential source of bioactive peptides. These major protein precursors have been found as protein in okara as a soybean by-product. Furthermore, primary structure, biological potential, and physicochemical, sensory, and allergenic characteristics of the theoretically released antioxidant peptides were predicted in this research. Glycinin and α subunits of β-conglycinin were selected as potential precursors of bioactive peptides based on in silico analysis. The most notable among these are antioxidant peptides. First, the potential of protein precursors for releasing bioactive peptides was evaluated by determining the frequency of occurrence of fragments with a given activity. Through the BIOPEP database analysis, there are several antioxidant bioactive peptides in glycinin and β and α subunits of β-conglycinin sequences. Then, an in silico proteolysis using selected enzymes (papain, bromelain) to obtain antioxidant peptides was investigated and then analyzed using PeptideRanker and Pepcalc. Allergenic analysis using the AllerTop revealed that all in silico proteolysis-derived antioxidant peptides are probably nonallergenic peptides. We also performed molecular docking against MPO (myeloperoxidases) for this peptide. Overall, the present study highlights that glycinin and β and α subunits of β-conglycinin could be promising precursors of bioactive peptides that have an antioxidant peptide for developing several applications.

## 1. Introduction

Almost all the world’s soybean production is used directly for human food [1]. Soybean has been commercialized in nature and soy-derived products include textured soy, soymilk, tofu, and fermented products (miso, shoyo, and tempeh). Soybean is a very important commodity in several areas as a protein source [1,2]. Okara is first developed by the production of soymilk. Soymilk can be made from whole SB or fat soy flour. Its production usually consists of five main stages: (1) SB washing to remove impurities; (2) SB soaking/hydration for 12 h at 25 °C, then draining and rinsing with water; (3) cooking at 98 °C for 5 min, with the aim of both sterilizing and improving aroma and nutritional value via the inactivation of trypsin inhibitors; (4) grinding in a blender with distilled water (1:10 ratio *w*/*v* SBs/water) for the preparation of a slurry; (5) separation of the slurry into soymilk (water-soluble SB extract) and okara by mechanical means (usually filtration) [3]. The production of soymilk made from soybean results in an insoluble by-product called okara with limited market value which is usually used as animal feed [4]. Okara is the waste produced during the production of soy milk, tofu, and soybean protein isolate in the food industry [5]. Okara has a high nutritive value due to its high-quality protein, fat, carbohydrates, and fiber that can still be utilized further. Concerning the proteins initially present in soybeans, approximately 23% are retained in the okara [3]. These proteins have a high nutritive value and superior protein efficiency ratio, indicating that okara is a potential source of low-cost vegetable protein for human consumption [6]. Besides protein, soybean and okara also contain bioactive compounds such as isoflavones [7].

In silico analysis is a common approach of bioinformatics that is widely used in analyzing the amino acid sequence, protein domain, and protein structure, through a computational approach [6,8,9]. In silico methods of peptide analyses could include different approaches such as homology modeling, molecular dynamics, protein docking, and PPI targeting. Structural characterization of the peptides could be carried out by x-ray crystallography, NMR spectroscopy, and cryo-electron microscopy instruments. The obtained structural data are stored and available in structural deposition databases like the Protein Data Bank (PDB). The advantages of computational in silico methods over empirical methods are their low cost, faster procedure speed, simple process, and reliability in target PPIs using peptides [10]. It is an advancement for the simulation of biological experiments in the early stage of research, especially for food proteins, to predict the proteins and peptide sequences, as well as their bioactivities [11,12]. In addition, potentially harmful substances that may be present within the proteins or peptides, such as toxins or allergens, can be discovered by using an in silico approach. Simulation of theoretical enzymatic hydrolysis, prediction of peptide sequences, and their potential biological activities can be conducted using the BIOPEP-UWM database, a freely accessible bioinformatics database that is widely used in research on biologically active peptides [13]. We already successfully applied this in silico approach analysis to identify bioactive peptides from several food sources such as yellowfin, tilapia, and jack bean [14,15]. 

Nowadays, the exploratory study of bioactive peptides from food proteins is gaining momentum, especially in finding functional properties that are beneficial for human health [16,17]. The bioactive peptides are commonly generated when their parent proteins are hydrolyzed by endogenous or exogenous enzymes into smaller fragments with their functional properties. Bioactive peptides derived from food proteins, containing 2–20 amino acid residues, have been reported to possess immunomodulatory, anti-cancer, anti-hypertensive, antioxidant, anti-proliferative, hypocholesterolemic, metal chelating, and anti-inflammatory activities [16,17]. For example, some health benefits of soybean bioactive peptides have been reported in various studies, including antioxidant activities [18]. They are potential food functional ingredients and can be used for different purposes, such as aroma enhancers, food additives, nutritional supplements, or therapeutic drugs. Much of the interest in plant-derived bioactive peptides present in foodstuffs lies in their potential pharmaceutical and/or nutraceutical benefits [19]. Well-known examples are soymetide (from soybean) and oryzatensin (from rice). In both cases, the bioactive peptide is a degradation product of an abundant protein present in the consumed part of the plant. The present bioinformatics or in silico approach has identified large numbers of proteins in food-producing plants from which bioactive peptides could be generated [19].

The antioxidant properties of soy protein have been ascribed to the cooperative effect of several properties including their ability to scavenge free radicals, act as a metal-ion chelator, oxygen quencher, or hydrogen donor, and the possibility of preventing the penetration of lipid oxidation initiators by forming a membrane around oil droplets [16]. The operational conditions employed in the processing of protein isolates, the type of protease, and the degree of hydrolysis affect the antioxidant and other biological activities [20]. The alternative substitution of synthetic antioxidants with natural ones is gaining interest due to consumer preferences and health concerns associated with the use of synthetic food additives [21]. Additionally, the importance of several of these natural antioxidants in the prevention or control of certain chronic diseases has been increasing. The use of proteins or their hydrolysates for food and/or cosmetic applications presents additional advantages over other antioxidants since they also confer nutritional and functional properties [22].

Glycinin (32 and 20 kDa respectively) and the β and α subunits of β-conglycinin (52 and 68 kDa) are major proteins in okara [23]. In this study, in silico analysis was performed on Glycinin and the β and α subunits of β-conglycinin to determine their effectiveness in releasing potential bioactive peptides with antioxidant activities using two different proteases, papain and bromelain, in BIOPEP-UWM database. We used two different proteases, papain and bromelain, in silico since these two proteases are mainly used and can be released into the bioactive peptides with antioxidant activities [24,25]. Previous research has already measured the physical treatment such as HPH and Alcalase to obtain the antioxidant peptide in Okara [5]. In our study, the antioxidant peptides will be analyzed to elucidate their physicochemical properties, gastrointestinal stability, toxicity, and allergenicity. In this study, we hypothesized that soy glycinin and the β and α subunits of β-conglycinin are proteins from which gastrointestinal-stable, non-toxic, and non-allergenic bioactive peptides can be liberated. The outcome of this study, although preliminary, could add to a better understanding of the potential of protein-rich food wastes, such as the soybean protein by-product okara, as a source of health-promoting peptides.

## 2. Results and Discussion

### 2.1. Profile of Potential Biological Activity

The antioxidant biological activities peptides released by glycinin and the β and α subunits of β-conglycinin are reported in Table 1. Hence, the time-saving and more economical computer-simulated method or in silico methods can be used to predict the generation of the bioactive peptides from glycinin and the β and α subunits of β-conglycinin. The biological activities of the glycinin and the β and α subunits of β-conglycinin were evaluated using the BIOPEP analysis tool. It was predicted by the tool that glycinin and the β and α subunits of β-conglycinin have the potential to release peptides with several antioxidant activities (Table 1). The antioxidant activities of bioactive peptides are mainly due to the presence of some aromatic amino acids, and histidine [11].

### 2.2. In Silico Proteolysis

The potential glycinin and the β and α subunits of β-conglycinin derived peptide sequences displaying the antioxidant peptide were predicted using the BIOPEP analysis tool. The lists of the theoretically released antioxidant peptide from glycinin and the β and α subunits of β-conglycinin are summarized in Table 2. The glycinin and the β and α subunits of β-conglycinin and protein sequences were also subjected to in silico proteolysis using the BIOPEP tool. The supposed antioxidant peptide derived from the hydrolysis of glycinin and the β and α subunits of β-conglycinin by papain and bromelain proteases are summarized and presented in (Table 2). 

The BIOPEP analysis method was used to predict possible glycinin and the β and α subunits of β-conglycinin-derived peptide sequences expressing the antioxidant activity. Table 2 summarizes the lists of antioxidant peptides released from glycinin as well as the β and α subunits of β-conglycinin using papain and bromelain enzymes. 

According to Table 2, the majority of peptides produced by hydrolysis with papain and bromelain are dipeptides. This finding is consistent with prior studies employing these two distinct enzymes but with a different raw material [25,26,27]. The antioxidant peptides produced by the enzyme combination were primarily dipeptides and tripeptides with antioxidant activity, as predicted by the BIOPEP tool. The combination of papain and bromelain can generate several antioxidant dipeptides and tripeptides.

### 2.3. Physiochemical Properties of Peptides

The antioxidant activity of peptides is affected by their structural characteristics, especially their molecular masses, amino acid composition, and peptide sequence [28]. Peptides composed of two to three amino acids have previously been shown to have a high antioxidant potential [29,30]. The calculated molecular weight of peptides derived from glycinin and conglycinin can be seen in Table 2. 

Furthermore, several properties of peptides are mentioned in Table 3. Peptides with low molecular weight are generally non-toxic and tend to be non-allergenic. The solubility, net charge, and iso-electric point are important physical parameters that are supposed to be considered in the design of novel drugs since these play a role in the distribution in the human body and in targeting specific bacteria, cells, proteins, or viruses. Consequently, further physical chemistry parameters of the peptides proposed were determined by the web server INNOVAGEN’s peptide calculator (PEPCALC) and the result is shown in Table 3. Peptides IR and EL were soluble in aqueous media. Moreover, identified peptides’ charges are varied (Table 3) at pH 7, which is a desirable property for MPO inhibitors (|charge| < −11), which is the net charge of MPO to avoid a possible cytotoxic effect.

### 2.4. Peptide Ranking of Antioxidant Peptides

The identified antioxidant peptide profile released from glycinin and conglycinin was subjected to activity prediction by in silico methods. PeptideRanker has been used to determine the antioxidant peptides’ potential. The PeptideRanker server can rank the peptide sets according to structure–function patterns [29]. The peptide generated from the papain and bromelain hydrolysis with the maximum peptide ranking scores can be seen in Table 4. The antioxidant dipeptides PHF and YYL showed the maximum peptide rank scores of 0.93 and 0.59, respectively (Table 4). Papain and bromelain mostly generate several active antioxidant peptides [2,14]. The toxicity of peptides derived from glycinin and conglycinin is one of the factors that can inhibit peptides’ development as functional food ingredients [31,32]. Based on Table 4, the sensory evaluation of the peptides that can be released from glycinin and conglycinin by papain and bromelain enzymes are mostly bitter and umami peptides. On the other hand, allergenic effect characteristics showed that most peptides can also cause allergenic effects. In silico toxicity and physicochemical predictions not only save time and money, but also facilitate the design of better therapeutic peptides, with low toxicity without compromising functionality [33,34]. Based on the result shown in Table 4, all the peptides derived from glycinin and conglycinin are not toxic.

### 2.5. Molecular Docking of the Peptide with the Antioxidant Enzyme In Silico (Binding Affinity and Interaction)

Molecular docking is an in silico method of mimicking the interaction model between receptors and small molecules [35]. Multiple molecular docking techniques were employed to choose MPO inhibitors with antioxidant activity, demonstrating the efficacy of this approach. Consequently, molecular docking is a helpful method to identify antioxidants and reveal underlying molecular mechanisms. Based on the energy binding affinity obtained through Pyrix, all the peptides derived from glycinin and conglycinin successfully docked to MPO spontaneously. Peptides derived from glycinin and conglycinin exhibit high affinity against MPO, and molecular interaction between peptides and MPO involved hydrogen bonds, van der walls bond, and several protein-ligand interactions (Figure 1). The presence of hydrogen bond interactions shows the stability of the complexes formed between peptide and protein. One of the most significant interactions between molecules is the hydrogen bond, which stabilizes the three-dimensional structure of proteins and nucleic acids. Moreover, all the peptides derived from glycinin and conglycinin show interaction with the important residue of MPO. The present finding confirmed the possibilities for peptides, especially dipeptides and tripeptides derived from glycinin and conglycinin, to act as antioxidants, hindering the substrate (such as H_2_O_2_) from accessing the heme-containing core of MPO. Compared to small compounds and proteins, peptides are much more flexible and do not have stable conformation before binding to a receptor [36]. To efficiently consider peptide flexibility in this study, we employed energy affinity binding dock the peptides derived from glycinin and conglycinin to MPO. The molecular docking result was expressed by (-) energy binding activity (Figure 2). The value of (-) energy indicates the bonding affinity between MPO and peptides, and a higher value means a more favorable combination [37].

The primary enzyme responsible for the production of ROS in vivo is MPO because heme peroxidase is known to occur in neutrophils and monocytes [34] MPO can produce highly active ROS (like HOCl) under peroxide H_2_O_2_ and halide conditions, leading to cell and tissue damage [34]. Consequently, molecular docking was performed to assess whether the various peptides were effective. According to Figure 2, the energy binding activity of peptides suggested that the peptide might spontaneously bind to the MPO.

Inhibitory peptides have the potential to inactivate an enzyme by occupying its active sites or obstructing access to the active site cavity of the enzyme [34]. As can be seen in Figure 3, the various peptides’ MPO docking postures were comparable to the earlier findings of antioxidant peptides from hemp seed, and both prevented entrance to the MPO active cavity, demonstrating their efficacy as antioxidants [33]. Hydrogen bonds are one sort of interaction between MPO and peptides. By preventing substances (such H_2_O_2_) from accessing the heme-containing core of MPO, peptides that attach to these locations can act as antioxidants in vivo. Therefore, like an earlier finding, their inhibitory effects on MPO may contribute to the antioxidant action of hemp seed protein-derived peptides in vivo [34].

## 3. Material and Methods

### 3.1. Profiles of the Potential Biological Activity

The BIOPEP analysis was used to evaluate the biological potential of glycinin and the β and α subunits of β-conglycinin (http://www.uwm.edu.pl/biochemia/index.php/en/biopep (accessed on 1 August 2022)). The sequences of glycinin and the β and α subunits of β-conglycinin obtained from the NCBI database were evaluated for potential biological activity profiles. The potential antioxidant peptide sequences of glycinin and the β and α subunits of β-conglycinin were also investigated. The frequency of occurrence of bioactive fragments in glycinin, as well as the β and α subunits of β-conglycinin, was also estimated. The sequences of glycinin and the β and α subunits of β-conglycinin were also subjected to in silico proteolysis to predict the theoretical peptide sequences cleaved by the enzymes papain and bromelain. Finally, a list of potential antioxidant peptides was compiled for further analysis.

### 3.2. Peptide Ranking

PeptideRanker (http://bioware.ucd.ie/compass/biowareweb/ (accessed on 1 August 2022)) was used to predict the ability of glycinin and the β and α subunits of β-conglycinin to generate antioxidant peptides. The peptide score of the chosen glycinin and conglycinin-derived antioxidant peptides was calculated using the PeptideRanker program. PeptideRanker assigned each peptide a score between 0 and 1. The highest score (1) indicated the most active peptides, while the lowest score (0) indicated the least active peptides.

### 3.3. Sensory Characteristics Prediction

Peptides and amino acids are examples of molecules that can modify the flavor of food. The BIOPEP analysis was used to predict the occurrence frequency of sensory features in hydrolyzed glycinin and the β and α subunits of β-conglycinin. The occurrence frequencies of the various sensory characteristics were also predicted for the generated peptides from glycinin and the β and α subunits of β-conglycinin using papain and bromelain. Furthermore, the sensory characteristics of the selected antioxidant were also predicted.

### 3.4. The Physicochemical Characteristics of the Antioxidant Peptides

Online peptide calculators were used to analyzing the potential antioxidant peptides. The online Pepcalc program (http://pepcalc.com/ (accessed on 1 August 2022)) was used to calculate the theoretical molecular weight, isoelectric point, peptide charge at pH 7, estimated solubility, and extinction coefficient of the screened antioxidative derived peptides from glycinin and β and α subunits of β-conglycinin peptides.

### 3.5. Allergenicity Prediction

One of the key problems in the development of soybean protein-derived functional food components is the allergenicity of the peptides. Therefore, AllerTop (https://ddg-pharmfac.net/AllerTOP/index.html (accessed on 1 August 2022)) was used to predict the allergenicity of the discovered glycinin and conglycinin-derived antioxidant peptides.

### 3.6. Molecular Docking against Peptide

AutoDock Vina- 4.2.6 was used to conducting a molecular docking investigation. The Crystal Structure of Human MPO (PDB ID: 3F9P) was retrieved from the RCSB Protein Data Bank (https://www.rcsb.org/ (accessed on 1 August 2022)), and the peptides derived were retrieved from PubChem. Both protein and peptides were prepared using PyRix. The activity was generated through the global docking method. 

### 3.7. Visualization

The visualization of docking results was carried out through PyMol (3D), and Biovia discovery (3D).

## 4. Conclusions

We can conclude that glycinin and conglycinin as major protein fractions in okara can be a precursor for several bioactive peptides with antioxidant activity using an in silico approach. The current study showed that using a papain and bromelain-based in silico method with glycinin and conglycinin chains as protein precursors successfully generated antioxidant peptides, the majority in the form of dipeptides and tripeptides. These results can be applied further for the use of a soybean by-product called okara as a precursor to antioxidant peptides in the development of functional food and nutraceuticals. Therefore, the use of okara, a by-product of soybean production, for producing bioactive peptides is in line with the application idea of valorization and can be viewed as a waste management strategy.

## Figures and Tables

**Figure 1 molecules-27-06855-f001:**
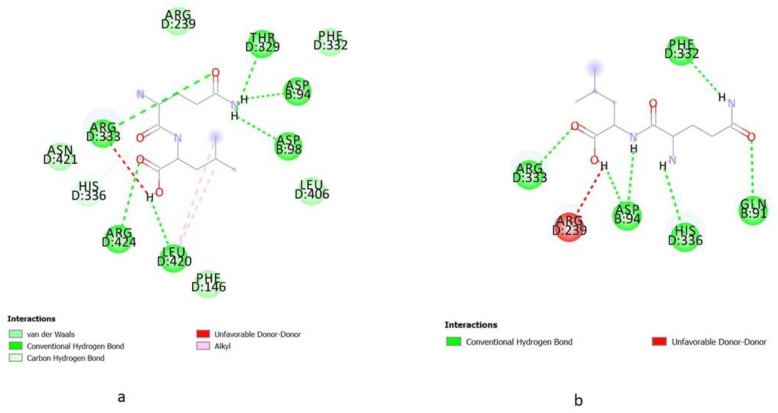
2D Ligand Protein interaction. (**a**) PHF; (**b**)YYL; (**c**) YNL; (**d**) HL; (**e**) IR; (**f**) YYV; (**g**) EL.

**Figure 2 molecules-27-06855-f002:**
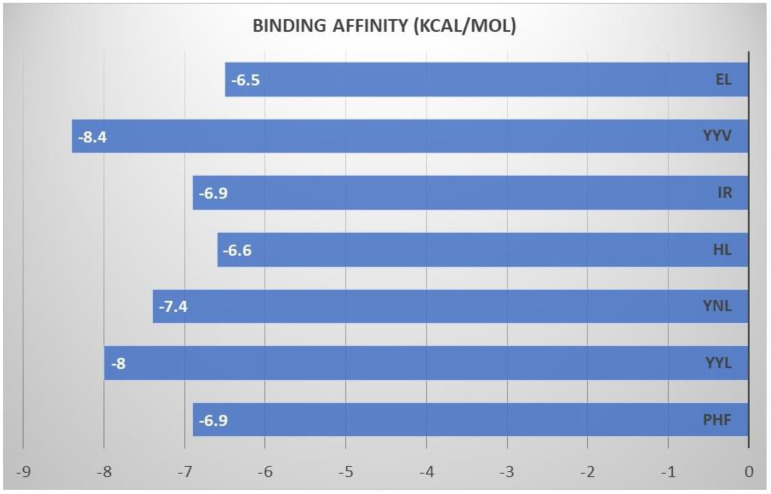
Binding free energy (binding activity) of ligand (peptides).

**Figure 3 molecules-27-06855-f003:**
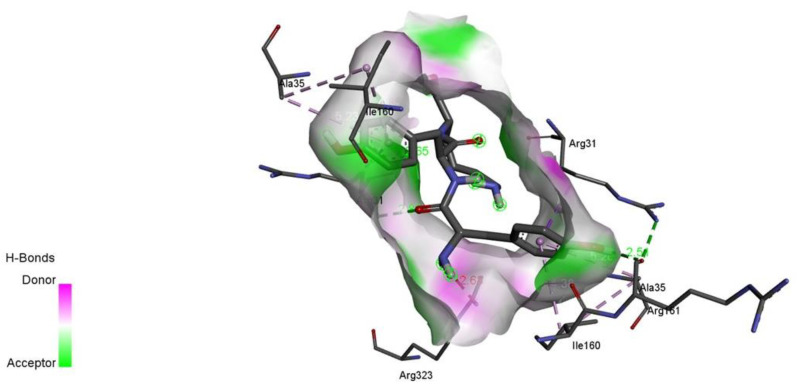
Example of 3D interaction of ligand-protein -YYV.

**Table 1 molecules-27-06855-t001:** Profile of biological activity as an antioxidative peptide of glycinin and to the β and α subunits of β-conglycinin.

No	Precursor	Activity	A	B
1	Glycinin	antioxidative	0.0543	
2	Conglycinin alpha subunit	antioxidative	0.0529	3.01898 × 10^−6^
3	Conglycinin beta subunit	antioxidative	0.0934	4.16056 × 10^−6^

**Table 2 molecules-27-06855-t002:** In silico assay using papain and bromelain release peptides with antioxidant activity.

No	Protein Precursor	Peptide ID	Sequence	Location	Monoisotopic Mass	Chemical Mass
1	Glycinin	3317	HL	(84–850	268.142	268.302
2		10019	YNL	(508–510)	408.1899	408.434
	beta conglycinin alpha subunit					
1		7888	EL	(190–191)	260.126	260.276
2		7888	EL	(513–514)	260.126	260.276
3		7942	YYV	(301–303)	443.194	443.473
4		8025	PHF	(465–467)	399.178	399.436
5		8215	IR	(220–221)	287.185	287.348
	beta conglycinin beta subunit					
1		7888	EL	(218–219)	260.126	260.276
2		7888	EL	(341–342)	260.126	260.276
3		7941	YYL	(134–136)	457.21	457.5
4		8025	PHF	(295–297)	399.178	399.436

**Table 3 molecules-27-06855-t003:** Physicochemical characterization of peptides derives from glycinin and conglycinin with antioxidant activities.

No	Sequence	Solubility	Net-Charge	Iso-Electric Point (pH)	Extinction Coefficient
1	PHF	Poor	0.1	8.26	0
2	YYL	Poor	0	3.32	2560 M^−1^cm^−1^
3	YNL	Poor	0	3.34	1280 M^−1^cm^−1^
4	HL	Poor	0.1	7.56	0
5	IR	Good	1	10.85	0
6	YYV	Poor	0	3.35	2560 M^−1^cm^−1^
7	EL	Good	−1	0.92	0

**Table 4 molecules-27-06855-t004:** PeptideRanker and sensory evaluation and allergenic prediction of peptides from glycinin and conglycinin as antioxidant peptides.

Number	Peptide	Peptide Ranker	Sensory Evaluation	Allergenicity	Toxicity
1	PHF	0.938016	bitter	Probable Non Allergen	Non Toxic
2	YYL	0.598987	bitter	Probable Non Allergen	Non Toxic
3	YNL	0.375314	bitter	Probable Non Allergen	Non Toxic
4	HL	0.374865	bitter	Probable Non Allergen	Non Toxic
5	IR	0.332363	bitter	Probable Non Allergen	Non Toxic
6	YYV	0.188713	bitter	Non Allergenic	Non Toxic
7	EL	0.0728272	umami	Probable Non Allergen	Non Toxic

## Data Availability

The data presented in this study are available on request from the corresponding author.

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
