# Peer review of "In Silico Approach of Glycinin and Conglycinin Chains of Soybean By-Product (Okara) Using Papain and Bromelain"

_molecules, 2022, doi:10.3390/molecules27206855_

Round 1

Reviewer 1 Report

The manuscript by Ningrum, A., described in silico approach to analysing proteins from soybean byproducts. Here the authors evaluate the potentials of glycinine and conglycinines as potential sources of bioactive peptides using different databases and in silico tools. Finally, they summarised the manuscript by stating that these proteins could be a promising precursor to bioactive peptides with antioxidant activity.

Overall, the manuscript is interesting and has a valuable contribution to the scientific field and society. But, in the current form, I still cannot recommend it for publication at Molecules. There are some concerns that I would like to raise:

1.    The title is not entirely clear. In silico approach is a computer method of analysing something. But in this case, it is directed to a noun. Therefore, I suggest the author clarify and change the title.

2.    The English language should be improved. At the current stage, reading the manuscript is not quite an enjoyable experience. There are some confusing, repetitive, and fragmented sentences.

3.    The authors performed molecular docking against MPO but did not define what MPO stand for.

4.    Many sentences need references, and the authors did not provide any citations. For example, the first sentence in the Introduction: “Approximately almost 10% of the world's soybean production is used directly for human food.”. I also found the authors lacked exploration of publications around this area. Several publications have been exploring the antioxidative activity of okara protein peptides, but the authors did not mention them. For example, Yokomizo, et al., 2002, Food Sci. Technol. Res., Fang, J., et al., 2021, Food Chem.:X., etc.

5.    The authors need to clarify the reference in this sentence: “Glycinin (32 and 20 kDa respectively) and the β and α subunits of β-conglycinin (52 and 68 kDa) are major proteins in okara [8].” The reference not specifically mentions okara’s major proteins. The publications that specifically study this is: Stanojevic, Sladjana P., et al. "Composition of proteins in okara as a byproduct in hydrothermal processing of soy milk." Journal of agricultural and food chemistry 60.36 (2012): 9221-9228.

Author Response

Dear Reviewer,

Thank you so much for your valuable input and suggestion for the manuscript.

Please kindly find the answer to your suggestions below :

My comments are the following:

The manuscript by Ningrum, A., described in silico approach to analysing proteins from soybean byproducts. Here the authors evaluate the potentials of glycinine and conglycinines as potential sources of bioactive peptides using different databases and in silico tools. Finally, they summarised the manuscript by stating that these proteins could be a promising precursor to bioactive peptides with antioxidant activity.

Overall, the manuscript is interesting and has a valuable contribution to the scientific field and society. But, in the current form, I still cannot recommend it for publication at Molecules. There are some concerns that I would like to raise:

  1. The title is not entirely clear. In silico approach is a computer method of analysing something. But in this case, it is directed to a noun. Therefore, I suggest the author clarify and change the title.

Answer:  Thank you so much for the suggestion.  We have revised the title based on the reviewer suggestion.

  1. The English language should be improved. At the current stage, reading the manuscript is not quite an enjoyable experience. There are some confusing, repetitive, and fragmented sentences.

Answer:  Thank you so much for the suggestion.  We tried our best to revise the manuscript

  1. The authors performed molecular docking against MPO but did not define what MPO stand for.

Answer:  Thank you so much for the suggestion .  We already added the definition of MPO that is stand for myeloperoxidases

  1. Many sentences need references, and the authors did not provide any citations. For example, the first sentence in the Introduction: “Approximately almost 10% of the world's soybean production is used directly for human food.”. I also found the authors lacked exploration of publications around this area. Several publications have been exploring the antioxidative activity of okara protein peptides, but the authors did not mention them. For example, Yokomizo, et al., 2002, Food Sci. Technol. Res., Fang, J., et al., 2021, Food Chem.:X., etc.

 Answer:  Thank you so much for the suggestion .  We already added the information on the introduction part.

  1. The authors need to clarify the reference in this sentence: “Glycinin (32 and 20 kDa respectively) and the β and α subunits of β-conglycinin (52 and 68 kDa) are major proteins in okara [8].” The reference not specifically mentions okara’s major proteins. The publications that specifically study this is: Stanojevic, Sladjana P., et al. "Composition of proteins in okara as a byproduct in hydrothermal processing of soy milk." Journal of agricultural and food chemistry60.36 (2012): 9221-9228.

 Answer:  Thank you so much for the suggestion .  We already revised the reference.

Thank you so much for your kind suggestions and advice.

We hope that our manuscript can be processed further in Molecules journal.

Kind regards,

Andriati Ningrum

Reviewer 2 Report

molecules-1970023: Although this paper is interesting and important, some concerns and several issues must be carefully revised.

1) Major concerns: You must check the plagiarism very carefully, and the citation of information in the manuscript must be seriously checked the correction. For example, at the first paragraph of Introduction, you just referred to Ref [1] only. In fact, this paragraph is copied from the first paragraph of Ref [1]. Although you cited Ref [1], those information were not found by Ref [1]. Those information came from (Riaz, 2006), (Liu, 2008), and (O’Toole, 1999), then Ref [1] referred to their works. Please seriously checked throughout the manuscript.

2) Major revision of the manuscript

2.1) Before jumping to second paragraph that mentioned about in silico, the overview of why the protein from soybean is important and are there any threat to soybean production or protein contain in soybean must be provided the information. For instance, the threats from climate change are not only impact to its production, but also the protein contain. Please see these papers. [Climate change impact on major crop yield and water footprint under CMIP6 climate projections in repeated drought and flood areas in Thailand. Science of the Total Environment 2022. 807, 150741.]  [Stability of Protein and Oil Content in Soybean across Dry and Normal Environments—A Case Study in Croatia. Agronomy 2022, 12, 915.]

 2.2) Figures 1 and 2 are poor resolution and very small font size. Please improve.

Author Response

Dear Reviewer,

Thank you so much for your valuable input and suggestion for the manuscript.

Please kindly find the answer to your suggestions below :

My comments are the following:

molecules-1970023: Although this paper is interesting and important, some concerns and several issues must be carefully revised.

1) Major concerns: You must check the plagiarism very carefully, and the citation of information in the manuscript must be seriously checked the correction. For example, at the first paragraph of Introduction, you just referred to Ref [1] only. In fact, this paragraph is copied from the first paragraph of Ref [1]. Although you cited Ref [1], those information were not found by Ref [1]. Those information came from (Riaz, 2006), (Liu, 2008), and (O’Toole, 1999), then Ref [1] referred to their works. Please seriously checked throughout the manuscript.

Answer:  Thank you so much for the suggestion.  We added the information

2) Major revision of the manuscript

2.1) Before jumping to second paragraph that mentioned about in silico, the overview of why the protein from soybean is important and are there any threat to soybean production or protein contain in soybean must be provided the information. For instance, the threats from climate change are not only impact to its production, but also the protein contain. Please see these papers. [Climate change impact on major crop yield and water footprint under CMIP6 climate projections in repeated drought and flood areas in Thailand. Science of the Total Environment 2022. 807, 150741.]  [Stability of Protein and Oil Content in Soybean across Dry and Normal Environments—A Case Study in Croatia. Agronomy 2022, 12, 915.]

Answer:  Thank you so much for the suggestion.  We added the information.

 2.2) Figures 1 and 2 are poor resolution and very small font size. Please improve.

Answer:  Thank you so much for the suggestion.  We already revised the manuscript

Thank you so much for your kind suggestions and advice.

We hope that our manuscript can be processed further in Molecules journal.

Kind regards,

Andriati Ningrum

Reviewer 3 Report

1-I would like to see the description of the food by-product used in the abstract introduction.

Okara is the waste produced during the production of soy milk, tofu, and soybean protein isolate in the food industry.

https://www.sciencedirect.com/science/article/pii/S2590157521000638

2-I would like to see the production flow of Okara.

https://www.ncbi.nlm.nih.gov/pmc/articles/PMC7248727/

3- In abstract, the time expressions seem to be wrong.

False:These major protein precur-sors were found as protein in okara as a soybean by-product.

True: These major protein precursors have been found as protein in okara as a soybean by-product.

There are books and helpful resources on time use in the academic article, or you can look at the published articles.

4-I think this information could have been given more appropriately. For those reading this for the first time, it would be better if these definitions were given in the first place. It can be sorted in an understandable way.

In silico methods of peptide analyses could include different approaches such as homology modeling, molecular dynamics, protein docking, and PPI targeting. Structural characterization of the peptides could be carried out by x-ray crystallography, NMR spectroscopy, and cryo-electron microscopy. The obtained structural data are stored and available in structural deposition databases like the Protein Data Bank (PDB). The advantages of computational in silico methods over empirical methods are their low cost, faster procedure speed, simple process, and reliability to target PPIs using peptides. This approach can lead to the atomic-level identification of PPIs (Murakami et al., 2017).

https://www.frontiersin.org/articles/10.3389/fmolb.2021.669431/full

5-Compounds of soy and by-product okara could also be given. If there are studies done, they could be added.

6-Bioactive peptide information and 2 main peptides in soy could also be given in more detail.

7-There seems to be a time error in the introduction. It will be good if the use of time in grammar, that is, information transfer, is reviewed.

8-I'm not sure how well this sentence fits scientifically.

‘Nowadays, the exploratory study of bioactive peptides from food proteins is gaining momentum, especially finding its functional properties that are beneficial for human health during this pandemic.’

9-This sentence also struck me as strange.

‘Although bioactive peptides are present in nature, they are commonly generated when their parent proteins are hydrolyzed by endogenous or exog-enous enzymes into smaller fragments with their functional properties.’

10-Isn't Figure 1 too small? I couldn't choose the ones given in the way.

11-Different studies on this subject could be cited as examples. If there is no similar study, it could be emphasized that your study is the first.

12-The title can also change to match the name of the journal (molecules).

‘In silico approach of glycinin and conglycinin chains of soybean by-product (okara) using papain and bromelain.’

Author Response

Dear Reviewer,

Thank you so much for your valuable input and suggestion for the manuscript.

Please kindly find the answer to your suggestions below :

My comments are the following:

1-I would like to see the description of the food by-product used in the abstract introduction.

Okara is the waste produced during the production of soy milk, tofu, and soybean protein isolate in the food industry.

https://www.sciencedirect.com/science/article/pii/S2590157521000638

 Answer:  Thank you so much for the suggestion .  We added the information.

2-I would like to see the production flow of Okara.

https://www.ncbi.nlm.nih.gov/pmc/articles/PMC7248727/

 Answer:  Thank you so much for the suggestion.  We added the information.

3- In abstract, the time expressions seem to be wrong.

False:These major protein precur-sors were found as protein in okara as a soybean by-product.

True: These major protein precursors have been found as protein in okara as a soybean by-product.

There are books and helpful resources on time use in the academic article, or you can look at the published articles.

 Answer:  Thank you so much for the suggestion.  We already revised it

4-I think this information could have been given more appropriately. For those reading this for the first time, it would be better if these definitions were given in the first place. It can be sorted in an understandable way.

In silico methods of peptide analyses could include different approaches such as homology modeling, molecular dynamics, protein docking, and PPI targeting. Structural characterization of the peptides could be carried out by x-ray crystallography, NMR spectroscopy, and cryo-electron microscopy. The obtained structural data are stored and available in structural deposition databases like the Protein Data Bank (PDB). The advantages of computational in silico methods over empirical methods are their low cost, faster procedure speed, simple process, and reliability to target PPIs using peptides. This approach can lead to the atomic-level identification of PPIs (Murakami et al., 2017).

https://www.frontiersin.org/articles/10.3389/fmolb.2021.669431/full

 Answer:  Thank you so much for the suggestion.  We already revised it

5-Compounds of soy and by-product okara could also be given. If there are studies done, they could be added.

Answer:  Thank you so much for the suggestion.  We added the information

6-Bioactive peptide information and 2 main peptides in soy could also be given in more detail.

Answer:  Thank you so much for the suggestion.  We added the information

7-There seems to be a time error in the introduction. It will be good if the use of time in grammar, that is, information transfer, is reviewed.

Answer:  Thank you so much for the suggestion.  We already revised it

8-I'm not sure how well this sentence fits scientifically.

‘Nowadays, the exploratory study of bioactive peptides from food proteins is gaining momentum, especially finding its functional properties that are beneficial for human health during this pandemic.’

Answer:  Thank you so much for the suggestion.  We already revised to :

Nowadays, the exploratory study of bioactive peptides from food proteins is gaining momentum, especially finding its functional properties that are beneficial for human health

9-This sentence also struck me as strange.

‘Although bioactive peptides are present in nature, they are commonly generated when their parent proteins are hydrolyzed by endogenous or exog-enous enzymes into smaller fragments with their functional properties.’

Answer:  Thank you so much for the suggestion.  We already revised to :

The bioactive peptides are commonly generated when their parent proteins are hydrolyzed by endogenous or exogenous enzymes into smaller fragments with their functional properties.’

10-Isn't Figure 1 too small? I couldn't choose the ones given in the way.

Answer:  Thank you so much for the suggestion.  We revised the Figure.  Hope it is better.

11-Different studies on this subject could be cited as examples. If there is no similar study, it could be emphasized that your study is the first.

Answer:  Thank you so much for the suggestion.  We added the information

12-The title can also change to match the name of the journal (molecules).

‘In silico approach of glycinin and conglycinin chains of soybean by-product (okara) using papain and bromelain.’

Answer:  Thank you so much for the suggestion.  We have revised the title based on the reviewer suggestion.

Thank you so much for your kind suggestions and advice.

We hope that our manuscript can be processed further in Molecules journal.

Kind regards,

Andriati Ningrum

Round 2

Reviewer 2 Report

Accept in present form.